# SAMRank: Unsupervised Keyphrase Extraction using Self-Attention Map in BERT and GPT-2

**Byungha Kang** and **Youhyun Shin**[*]
Department of Computer Science and Engineering, Incheon National University
{bhkang, yhshin}@inu.ac.kr

## Abstract

We propose a novel unsupervised keyphrase extraction approach, called SAMRank, which uses only a self-attention map in a pretrained language model (PLM) to determine the importance of phrases. Most recent approaches for unsupervised keyphrase extraction mainly utilize contextualized embeddings to capture semantic relevance between words, sentences, and documents. However, due to the anisotropic nature of contextual embeddings, these approaches may not be optimal for semantic similarity measurements. SAMRank as proposed here computes the importance of phrases solely leveraging a self-attention map in a PLM, in this case BERT and GPT-2, eliminating the need to measure embedding similarities. To assess the level of importance, SAMRank combines both global and proportional attention scores through calculations using a self-attention map. We evaluate the SAMRank on three keyphrase extraction datasets: Inspec, SemEval2010, and SemEval2017. The experimental results show that SAMRank outperforms most embedding-based models on both long and short documents and demonstrating that it is possible to use only a self-attention map for keyphrase extraction without relying on embeddings. Source code is available at https://github.com/kangnlp/SAMRank.

## 1 Introduction

Keyphrase extraction refers to process of identifying the words or phrases that signify the primary themes of a document. It has a wide range of applications, including document summarization, information retrieval, and topic modeling. The methodologies used for keyphrase extraction are typically categorized into supervised and unsupervised approaches. Although supervised keyphrase extraction yields excellent performance, it relies heavily on large quantities of labeled data and tends to be domain-specific. In contrast, unsupervised keyphrase extraction uses only the information intrinsic to the document. Traditional methods rely on statistical information, such as TF-IDF (Jones, 2004), and graphs based on the co-occurrence of words (Mihalcea and Tarau, 2004). However, these methods often fall short of deciphering latent meanings in the text, resulting in suboptimal performance. The introduction of deep learning-based language models changed this paradigm, by enabling the extraction of phrases closely connected semantically to the document through the calculation of the degree of similarity between the document and phrase embeddings.

However, this paper does not utilize embeddings. The rationale behind this is that contextualized representations are not well-suited to tasks based on similarity. A study that has geometrically analyzed the representations of PLMs such as ELMo (Peters et al., 2018), BERT (Devlin et al., 2019), and GPT-2 (Radford et al., 2019), have postulated that the contextualized representations of these language models exhibit anisotropy (Ethayarajh, 2019). These representations tend to concentrate on specific directions of the vector space, becoming increasingly anisotropic in higher layers. Indeed, the average cosine similarity of two word representations randomly chosen from BERT and GPT-2 markedly increases with the layer depth. Given that embeddings typically come from the last layer's representation of a PLM, anisotropic embeddings can compromise tasks that require similarity-based rankings. To counteract this, BWRank (Wang and Li, 2022) improved the performance of keyphrase extraction by isotropically reconstructing BERT's embeddings by reducing dimensions through whitening operation.

In an earlier work entitled "What Does BERT Look At? An Analysis of BERT's Attention" (Clark et al., 2019), the authors sought to understand how BERT's attention mechanism oper-

---

[*]Corresponding author.

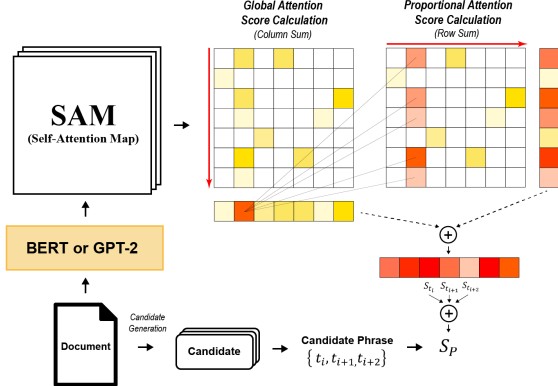

Figure 1: Overview of SAMRank.

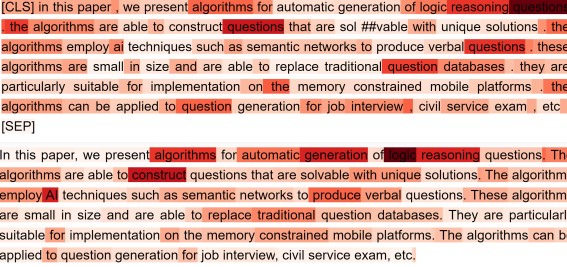

Figure 2: Visualization of final token-level scores. The top figure uses the self-attention map from BERT (2nd head of the 5th layer), and the bottom one from GPT-2 (1st head of the 11th layer). Deeper colors indicate higher scores.

ates by determining how much attention a specific word gives to other words in each head of BERT. Their findings indicate that certain BERT heads were skilled in identifying grammatical structures and coreferences such as direct objects of verbs and objects of prepositions. This suggests that in transformer-based PLMs, such as BERT that employ a multi-head attention mechanism to handle linguistic complexity, certain heads may specialize in particular tasks. Building on this research, we can posit that some heads of a transformer-based PLM may specialize in capturing keyphrases given their importance in providing an understanding of the thematic essence of the text and considering the likelihood that some heads may have been trained to detect keyphrases during the pre-training process of the PLM.

Therefore, this paper shifts the focus from PLM embeddings to the self-attention map afforded by each head in the PLM. We propose SAMRank, a novel approach that leverages only the self-attention map of a transformer-based PLM to extract keyphrases. SAMRank calculates a global attention score by summing the attention weights received by the token from all tokens within the document. It also redistributes the global attention score proportionally to other tokens that are focusing on globally significant tokens based on the attention weights they have allocated, yielding a proportional attention score. The combination of these two scores determines the final score for tokens. By aggregating the final scores of each token within a candidate phrase, SAMRank calculates the total score for the phrase and determines its overall significance. This results in ranking of candidate phrases based on their total scores, thus allowing the extraction of phrases that both receive high

attention and are highly attentive to phrases receiving global attention. An overview of SAMRank is presented in Figure 1 and the final token scores calculated through SAMRank can be visualized, as depicted in Figure 2.

In this paper, we independently analyze the self-attention map of each head in each layer to identify heads that excel in keyphrase extraction. We extract the self-attention map from all 144 heads in BERT and GPT-2, using three representative keyphrase extraction datasets: Inspec, SemEval2010, and SemEval2017. The performance of each head is evaluated, with experimental results showing that SAMRank outperforms most embedding-based methods in specific heads of both BERT and GPT-2.

This paper's main contributions are three-fold:

- We propose SAMRank, a novel and efficient approach that extracts keyphrases solely with self-attention map, eliminating the need for embeddings and similarity measurements.

- We demonstrate not only the capability of encoder-based language models like BERT, but also decoder-based language models such as GPT-2, for keyphrase extraction. Furthermore, we provide evidence that GPT-2 can outperform BERT in keyphrase extraction tasks.

- We discover the existence of heads in transformer-based PLMs that are more proficient in keyphrase extraction compared to baselines.

## 2 Background

The transformer model proposed in "Attention Is All You Need" (Vaswani et al., 2017) introduced the attention mechanism, bringing about a revolution in the field of natural language processing.

In this section, we introduce a self-attention map, which is the computational output of self-attention, the core mechanism of the transformer. Additionally, we discuss two transformer-based PLMs, BERT and GPT-2, which have demonstrated superior performance in various natural language processing tasks.

## 2.1 Self-Attention Map

The transformer's pivotal component is the self-attention mechanism, which models intricate interdependencies among tokens within a text by quantifying the attention each token pays to the others. The self-attention mechanism of the transformer calculates the scaled dot product between each token's query vector ($Q$) and the key vectors ($K$s) of all tokens, then passes this through a softmax function to compute the attention weights. The attention weights are multiplied by each token's value vector ($V$), and these vectors are summed to generate the final output vector for the query token.

Since each token in a text is served as a query $Q$ once in self-attention, the aggregation of the attention weights of all tokens yields a square matrix, known as the self-attention map (SAM). SAM is a crucial piece of information that provides insights into how tokens relate to each other. The i-th row of SAM represents the attention weights of the i-th token of the text when it is served as a query $Q$—meaning, how much it 'attends to' the other tokens—with the sum of each row equaling 1. Conversely, the i-th column of SAM represents the attention weights received by the i-th token from the other tokens when it is served as a value $K$.

While self-attention considers the relationships with all tokens in the input text, masked self-attention 'masks' or disregards tokens that appear after the current token. This means that each token pays attention only to the tokens that appeared before it. Consequently, a masked self-attention map is a lower triangular matrix where all values above the diagonal are 0.

Transformer is an encoder-decoder model, with the encoder employing the self-attention mechanism and the decoder using the masked self-attention mechanism. The transformer is multi-layered, with the attention mechanism operating in parallel across multiple heads within each layer. Each head has independent $Q$, $K$, and $V$ weights, meaning they process the same input text from various perspectives. The outputs from each head are concatenated to form the final layer output.

## 2.2 BERT

BERT (Devlin et al., 2019) is a model based on the encoder of the transformer, which utilizes self-attention, resulting in a square matrix for each head's self-attention map. In this paper, we use the BERT-base model[1], which comprises 12 layers, each equipped with 12 attention heads, yielding a total of 144 heads. The maximum input token length for BERT-base is 512.

## 2.3 GPT-2

GPT-2 (Radford et al., 2019) is a model based on the decoder of the transformer, which uses masked self-attention, creating a lower triangular matrix for each head's self-attention map. In this paper, we use the 117M size model of GPT-2[2]. The GPT-2 model, consists of 12 layers, each with 12 attention heads, totaling 144 heads. The maximum input token length for GPT-2 is 1024.

## 3 Methodology

Our proposed methodology involves ranking phrases based on the self-attention map (SAM) of transformer-based language models, in this case BERT and GPT-2. Each row of SAM, which is a self-attention weights vector, is perceived as each token allotting its attention according to its relevance with other tokens. The tokens that are semantically connected to many other tokens are more likely to receive this 'attention'. Hence, the global attention score of each token is computed by summing the attention weight it received from all the tokens in the text. Then, similar to the attention operation (multiplying the $V$ vector by the attention weight), the global attention score is distributed in proportion to the attention weight assigned by the token. This results in the calculation of the proportional attention score, where tokens strongly associated with globally significant tokens are assigned high scores. The final score for a token is calculated by combining the global attention score and the proportional attention score. For phrases, their scores are computed as the accumulated sum of the scores of their constituent tokens across all occurrences in the document. Based on these phrase-level scores, the phrases are then ranked to determine their overall importance. Table 1 provides an instance of

---

[1] https://huggingface.co/bert-base-uncased
[2] https://huggingface.co/gpt2

these extraction results for a specific text.

## 3.1 Candidate Generation

We use the module implemented in EmbedRank (Bennani-Smires et al., 2018) to extract keyphrase candidates. This module employs the Stanford-CoreNLP tool[3] to tokenize and POS tag the documents and then uses NLTK[4]'s RegexpParser to extract noun phrases, which are defined as words or series of words tagged with 'NN', 'NNS', 'NNP', 'NNPS', and 'JJ'. These extracted noun phrases become candidate phrases.

## 3.2 Self-Attention Map Extraction

In this paper, we extract self-attention maps (SAMs) from each layer and head of BERT and GPT-2 using Hugging Face's transformers library[5]. We input the entire document into the PLM at once, as opposed to inputting each sentence separately. This enables us to capture the correlations between tokens that reflect the overall context of the document. If the number of tokens in the document exceeds the model's maximum number of input tokens, we divide the entire document into several segments with the same number of tokens and extract SAM for each. For BERT, we input text without [PAD] tokens in the structure of the [CLS] text [SEP]. GPT-2 is input directly without any special tokens.

## 3.3 Global Attention Score Calculation

SAMRank initially calculates the score at the token-level and later aggregates it at the phrase-level.

We calculate each token's global attention score as the total sum of attention weights it received from all the tokens in the document. The self-attention map is a matrix $A$, its size determined as follows: input token number ($n$) × input token number ($n$). Therefore, we can calculate each token's global attention score ($G_{t_i}$) by determining the column sum for each column of SAM.

$$A = [A]_{n \times n} \quad (1)$$

$$G_{t_i} = \sum_{j=1}^{n} A_{ji} \quad (2)$$

However, BERT's [SEP] tokens receive very high attention weights from other tokens from

[3] https://stanfordnlp.github.io/CoreNLP
[4] https://github.com/nltk
[5] https://github.com/huggingface/transformers

the middle layer (Clark et al., 2019). Also, the first token of GPT-2, which performs masked self-attention, receives very high attention weights from the beginning of the document. We set the global attention score for BERT's [SEP] tokens and GPT-2's first input tokens to 0 because their global attention scores can be excessively high, which can hinder the understanding of correlation between each token.

## 3.4 Proportional Attention Score Calculation

If a token is not globally important but has a very high correlation with an important token, the token should also be considered an important token. If a token assigns a very high attention weight to an important token, it can be interpreted as having a high correlation with the important token. We allow such tokens to be ranked high by redistributing the global attention score in proportion to the attention weight they assigned to the important token.

$$B = A \cdot \text{diag}(G) \quad (3)$$

This can be calculated by multiplying each token's global attention score by each element in the corresponding column of SAM.

$$B'_{ji} = \frac{B_{ji}}{\sum_{j=1}^{n} B_{ji}} \quad (4)$$

Then, we normalize the values of each column by dividing them by the sum of the column to make each column have values between 0 and 1, thus obtaining matrix $B'$.

$$P_{t_i} = \sum_{j=1}^{n} B'_{ij} \quad (5)$$

The proportional attention score, which is calculated proportional to each token's global importance and its attention weight, is calculated by performing a row sum operation for each row of the updated SAM ($B'$).

## 3.5 Phrase-level Score Aggregation

The final score at the token-level is determined by the sum of the global attention score ($G_{t_i}$) and the proportional attention score ($P_{t_i}$) of each token:

$$S_{t_i} = G_{t_i} + P_{t_i} \quad (6)$$

The score at the phrase-level is calculated as the sum of the final importance scores of the tokens

that make up the phrase:

$$S_{P_k} = \sum_{i \in P_k} S_{t_i} \quad (7)$$

The final score at the phrase-level is calculated as the sum of the phrase scores calculated at each location where the phrase appears in the document:

$$S_P = \sum_{k \in P} S_{P_k} \quad (8)$$

If a phrase consists of a single word, and if that word is used as a subword of another candidate phrase, the score at that location is not included in the calculation because the score of the subword is calculated in the context of the phrase to which the subword belongs, and this score should be added to the score of the phrase. Therefore, a single word is only scored at positions where it is used independently and not as a subword of other candidate phrases. Also, single words are often most likely to receive higher scores simply because they are more frequent in the document than other candidate phrases. To correct this, the final score of a single word is calculated as the average score by dividing the total score of the word by the frequency of the word.

For long documents that exceed the maximum number of input tokens, first we divide them into multiple segments and apply SAMRank independently to each segment, after which we integrate each result. That is, the final score of the phrase is ranked by calculating it as the sum of the scores that the phrase obtained in each segment.

## 4 Experiments

### 4.1 Datasets and Evaluation Metrics

We evaluate the proposed method using three commonly used datasets: Inspec (Hulth, 2003), SemEval2010 (Kim et al., 2010), and SemEval2017 (Augenstein et al., 2017). Both Inspec and SemEval2017 datasets consist of scientific paper abstracts, and no document in these datasets exceeds BERT's maximum input length of 512 tokens. In contrast, the SemEval2010 dataset comprises full-length ACM papers, most of which exceed GPT-2's maximum input token count of 1024. In line with previous studies (Saxena et al., 2020; Liang et al., 2021), we use the test set composed of 100 documents and their annotated set of keyphrases. Table 2 presents the statistical information of each dataset.

| Teaching management science with spreadsheets: From decision models to decision support. The 1990s were a decade of enormous change for management science (MS) educators. While the outlook at the beginning of the decade was somewhat bleak, the renaissance in MS education brought about by the use of spreadsheets as the primary delivery vehicle for quantitative modeling techniques has resulted in a much brighter future. This paper takes inventory of the current state of MS education and suggests some promising new directions in the area of decision support systems for MS educators to consider for the future. |
|---|
| **Ground Truth** — management science; spreadsheets; quantitative modeling; MS education; decision support systems |
| **JointGL** — management science; decision models; enormous change; MS education; decision support |
| **SAMRank (BERT)** — spreadsheets; quantitative modeling techniques; management science; decision support; decision support systems |
| **SAMRank (GPT-2)** — spreadsheets; management science; promising new directions; decision support systems; decision support |

Table 1: An example of top 5 keyphrase extraction results by JointGL and SAMRank from the Inspec dataset.

| Datasets | Inspec | SemEval2010 | SemEval2017 |
|---|---|---|---|
| Docs | 500 | 100 | 493 |
| Avg. Words | 135 | 1589 | 194 |
| Avg. Sents | 6 | 68 | 7 |
| Avg. Keys | 9 | 12 | 17 |
| Unigram | 13.47% | 19.52% | 24.59% |
| Bigram | 52.66% | 54.57% | 33.61% |
| Trigram | 24.86% | 19.02% | 17.40% |

Table 2: Statistics of datasets.

To evaluate the performance of keyphrase extraction, we calculate the F1 measure for the top 5, top 10, and top 15 keyphrases predicted by the model. This is done after ranking the predicted keyphrases, applying NLTK's PorterStemmer for stemming, and eliminating duplicates.

### 4.2 Baselines and Implementation Details

We compare SAMRank with statistics-based methods: TF-IDF (Jones, 2004), YAKE (Campos et al., 2018); the graph-based methods: TextRank (Mihalcea and Tarau, 2004), SingleRank (Wan and Xiao, 2008), TopicRank (Bougouin et al., 2013), PositionRank (Florescu and Caragea, 2017); static embedding-based methods: EmbedRank (Bennani-Smires et al., 2018), and contextual embedding-based methods: SIFRank (Sun et al., 2020), AttentionRank (Ding and Luo, 2021), MDERank (Zhang et al., 2022), JointGL (Liang et al., 2021). Most of these methods evaluate performance outcomes by stemming the predicted phrases and the ground truth phrases and then removing duplicates. However, JointGL, which shows the highest performance among the existing methods, does not remove duplicates after stemming. This may have inflated the reported performance. To ensure a fair comparison, we reproduce JointGL's performance by removing duplicates after stemming. We use the optimal hyperparameters reported in the JointGL paper. For SemEval2017, we use the same hyperparameters used with Inspec.

| Model | Inspec | | | SemEval2010 | | | SemEval2017 | | |
|---|---|---|---|---|---|---|---|---|---|
| | F1@5 | F1@10 | F1@15 | F1@5 | F1@10 | F1@15 | F1@5 | F1@10 | F1@15 |
| Statistics-based Models | | | | | | | | | |
| TF-IDF | 11.28 | 13.88 | 13.83 | 2.81 | 3.48 | 3.91 | 12.70 | 16.26 | 16.73 |
| YAKE | 18.08 | 19.62 | 20.11 | 11.76 | 14.4 | 15.19 | 11.84 | 18.14 | 20.55 |
| Graph-based Models | | | | | | | | | |
| TextRank | 27.04 | 25.08 | 36.65 | 3.80 | 5.38 | 7.65 | 16.43 | 25.83 | 30.50 |
| SingleRank | 27.79 | 34.46 | 36.05 | 5.90 | 9.02 | 10.58 | 18.23 | 27.73 | 31.73 |
| TopicRnak | 25.38 | 28.46 | 29.49 | 12.12 | 12.90 | 13.54 | 17.10 | 22.62 | 24.87 |
| PositionRank | 28.12 | 32.87 | 33.32 | 9.84 | 13.34 | 14.33 | 18.23 | 26.30 | 30.55 |
| Static Embedding-based Models | | | | | | | | | |
| EmbedRank d2v | 31.51 | 37.94 | 37.96 | 3.02 | 5.08 | 7.23 | 20.21 | 29.59 | 33.94 |
| EmbedRank s2v | 29.88 | 37.09 | 38.40 | 5.40 | 8.91 | 10.06 | - | - | - |
| Contextual Embedding-based Models | | | | | | | | | |
| SIFRank | 29.11 | 38.80 | 39.59 | - | - | - | 22.59 | 32.85 | 38.10 |
| AttentionRank | 24.45 | 32.15 | 34.49 | 11.39 | 15.12 | 16.66 | 23.59 | 34.37 | 38.21 |
| MDERank | 27.85 | 34.36 | 36.40 | 13.05 | 18.27 | 20.35 | 20.37 | 31.21 | 36.63 |
| JointGL | 30.82 | 36.28 | 36.67 | 10.78 | 13.67 | 14.64 | 20.49 | 29.63 | 34.05 |
| Self-Attention Map-based Models (Ours) | | | | | | | | | |
| SAMRank (BERT) | **33.96** | **39.35** | **39.73** | **15.28** | **18.36** | 18.03 | **24.08** | 33.40 | 37.53 |
| SAMRank (GPT-2) | **33.92** | **39.44** | **39.72** | **15.88** | **19.49** | 19.03 | **24.80** | **34.75** | **38.78** |

Table 3: Performance comparison of baselines and SAMRank on F1@5, F1@10, and F1@15. The **bold** indicates cases where SAMRank showed improved performance compared to the baseline models. The underline indicates the best performance model.

## 4.3 Results

In Table 3, the experimental results for SAMRank are presented. We select the head that achieved the highest F1@15 score from a total of 144 heads. In Inspec, the eleventh head of the third layer of BERT and the first head of the eleventh layer of GPT-2 performed best. Both SAMRank using BERT and GPT-2 surpass the F1@5, F1@10, and F1@15 scores of existing baselines. In SemEval2010, the sixth head of the twelfth layer of BERT and the eleventh head of the eleventh layer of GPT-2 obtain the highest performance. Our model achieves state-of-the-art (SOTA) performance on F1@5, F1@10, and outperforms all other baselines on F1@15, except for MDERank. These results demonstrate that SAMRank also performs well when extracting keyphrases from long documents. In SemEval2017, the second head of the fifth layer of BERT and the first head of the eleventh layer of GPT-2 show the highest performance, achieving SOTA results on F1@5, F1@10, and F1@15.

Our proposed SAMRank outperforms almost all existing baselines across all datasets. We note that the performance of GPT-2 is slightly higher than BERT, except for the Inspec dataset. An earlier work that analyzed the embeddings of BERT and GPT-2 pointed out that the average intra-sentence similarity, the cosine similarity between each word representation within a sentence in GPT-2, is lower than it is in BERT (Ethayarajh, 2019). We also conjecture that while BERT encodes the context by making each word in a sentence share meaning, GPT-2 maintains the unique meaning of each word while encoding the context. Therefore, it can be hypothesized that the characteristic of GPT-2, which preserves the unique meaning of each word during contextualization without dilution, worked favorably, resulting in GPT-2 showing higher performance than BERT.

## 4.4 Ablation Study

### 4.4.1 Contribution of Global and Proportional Attention Score

Table 4 presents the performance evaluation when the global attention score and proportional attention score, components of SAMRank, are used independently for the head showing the highest F1@15 performance. Even when using only the global attention score, both BERT and GPT-2 showed fairly high performance outcomes across all datasets. However, the F1@5, F1@10, and F1@15 scores are approximately 1% lower on average across all

| Model | Inspec | | | SemEval2010 | | | SemEval2017 | | |
|---|---|---|---|---|---|---|---|---|---|
| | F1@5 | F1@10 | F1@15 | F1@5 | F1@10 | F1@15 | F1@5 | F1@10 | F1@15 |
| SAMRank (BERT) | **33.96** | **39.35** | **39.73** | **15.28** | **18.36** | **18.03** | **24.08** | **33.40** | **37.53** |
| only Global | 33.43 | 38.04 | 38.36 | 14.95 | 16.26 | 17.41 | 23.39 | 33.29 | 36.89 |
| only Proportional | 32.63 | 38.54 | 39.04 | 14.77 | 17.65 | 17.39 | 23.46 | 32.98 | 36.89 |
| SAMRank (GPT-2) | 33.92 | **39.44** | 39.72 | **15.88** | **19.49** | **19.03** | **24.80** | **34.75** | **38.78** |
| only Global | 33.66 | 39.24 | 38.33 | 14.96 | 18.01 | 18.76 | 24.07 | 33.82 | 37.13 |
| only Proportional | 25.02 | 32.74 | 34.23 | 11.27 | 14.29 | 14.87 | 19.76 | 28.72 | 32.98 |

Table 4: Experimental results on the impact of each attention score: global and proportional attention score. Comparison of performances using only the global attention score (only Global) and only the proportional attention score (only Proportional). The **bold** indicates the best performance within each group (BERT and GPT-2).

| Model | Inspec | | | SemEval2010 | | | SemEval2017 | | |
|---|---|---|---|---|---|---|---|---|---|
| | F1@5 | F1@10 | F1@15 | F1@5 | F1@10 | F1@15 | F1@5 | F1@10 | F1@15 |
| Backward Redistribution | | | | | | | | | |
| SAMRank (BERT) | **33.96** | 39.35 | 39.73 | 15.28 | 18.36 | 18.03 | **24.08** | **33.40** | **37.53** |
| SAMRank (GPT-2) | **33.92** | **39.44** | **39.72** | 15.88 | **19.49** | **19.03** | 24.80 | **34.75** | **38.78** |
| Forward Redistribution | | | | | | | | | |
| SAMRank (BERT) | **33.96** | **39.67** | **39.78** | **16.08** | **18.77** | **18.27** | 23.04 | 32.49 | 37.18 |
| SAMRank (GPT-2) | 33.50 | 38.80 | 39.04 | **17.17** | 18.40 | 18.90 | **25.18** | 34.60 | 38.22 |

Table 5: Experimental results on the comparison of global attention score redistribution directions for computing the proportional attention score. Evaluation with different distribution directions, both backward and forward.

datasets compared to when both the global and proportional attention scores are used. Additionally, performance degradation is more evident when only the proportional attention score is used. Especially for SAMRank (GPT-2), when only the proportional attention score was used, the F1@5, F1@10, and F1@15 scores decreased by an average of approximately 8% across all datasets. These results indicate that the global attention score is a factor that more crucially affects the performance of SAMRank. However, as the best performance was found when both the proportional attention score and global attention score were used in all cases, both scores are essential in SAMRank, and a more accurate keyphrase extraction is possible by combining them.

### 4.4.2 Direction of the Global Attention Score Redistribution

Table 5 presents the results of the experiments for both the backward and forward approaches. In Section 3.4, we propose a method of redistributing the global attention score in proportion to the attention weight each token receives (the backward approach). Conversely, each token could also redistribute the global attention score in proportion to the attention weights it allocates to other tokens as a query $Q$ (the forward approach). We conduct experiments with the forward approach as well. The results show that different heads exhibit the highest performance in the forward approach and the backward approach. SAMRank based on BERT heads performs better with the forward approach on the Inspec and SemEval2010 datasets. However, on the SemEval2017 dataset, the original backward approach achieves better performance. On the other hand, SAMRank based on GPT-2 heads mostly shows better performance with the original backward approach, with the forward approach only showing better performance on F1@5 of the SemEval2010 and SemEval2017 datasets. Therefore, the performance based on the distribution direction of the global attention score varied depending on the dataset and the model.

We conclude that the backward and forward approaches evaluate the importance of tokens from different perspectives. However, the average performance difference based on the distribution direction is not significant. This shows that the overall approach of SAMRank, which calculates the global attention score and redistributes it based on the relevance of each token individually, is effective.

## 5 Analysis of Heads

To identify heads that specialized in keyphrase extraction, we evaluate the performance of each of

| PLM | Inspec | | | SemEval2010 | | | SemEval2017 | | |
|---|---|---|---|---|---|---|---|---|---|
| | F1@5 | F1@10 | F1@15 | F1@5 | F1@10 | F1@15 | F1@5 | F1@10 | F1@15 |
| BERT | 33.96 | 39.35 | 39.73 | 15.28 | 18.36 | 18.03 | 24.08 | 33.40 | 37.53 |
| GPT-2 | 33.92 | 39.44 | 39.72 | 15.88 | **19.49** | 19.03 | 24.80 | 34.75 | **38.78** |
| **Llama 2 7B** | 33.28 | 39.57 | 39.27 | 16.69 | 18.66 | 18.51 | 23.22 | 33.38 | 37.63 |
| **Llama 2 13B** | 34.07 | 39.75 | 39.51 | 16.11 | 18.47 | **19.12** | 23.94 | 33.97 | 38.04 |
| **Llama 2 70B** | **35.06** | **40.01** | **39.76** | **17.03** | 19.02 | 18.90 | **24.88** | **35.32** | 38.30 |

Table 6: Performance of SAMRank on LLMs: Llama 2 7B, Llama 2 13B, and Llama 2 70B.

the 144 heads in BERT and GPT-2. The F1@15 performance of each head across all layers for the three datasets is detailed in Appendix A. We find that the performance of specific heads shows consistent patterns across all three datasets. In BERT, heads mainly located in the third and fifth layers demonstrate high performance, whereas in GPT-2, heads in the eleventh layer are particularly performant. However, the head yielding the highest performance varies across the datasets. This indicates that specific heads in transformer-based PLMs do pay substantial attention to phrases perceived as keyphrases by humans, but their performance varies depending on the type and length of the text, suggesting that each head might adopt different perspectives or attention patterns when identifying keyphrases.

Moreover, we observe that the intra-layer performance variance in BERT is larger than in GPT-2. This might suggest that some heads in BERT specialize in tasks unrelated to semantic keyphrase extraction, possibly focusing on syntactic or other non-semantic tasks. Despite the relatively small performance differences among the top-performing heads across different layers in BERT, we identify some particular heads (first and eleventh head) in the eleventh layer of GPT-2 that significantly outperforms all other heads, including those in BERT. This suggests the possibility of certain heads in GPT-2 being highly specialized for keyphrase identification.

## 6 SAMRank on LLMs

SAMRank is based on the self-attention mechanism, so it can be used in transformer models other than BERT or GPT-2. We conduct keyphrase extraction experiments applying SAMRank on the large language models (LLMs) that have recently attracted significant attention in the field of natural language processing. We utilize Llama 2, recently released by Meta (Touvron et al., 2023). We per-

form experiments on: Llama 2 7B [6], Llama 2 13B [7], and Llama 2 70B [8]. Llama 2 7B consists of 32 layers, each with 32 heads. Llama 2 13B consists of 40 layers, each with 40 heads. Llama 2 70B consists of 80 layers, each with 64 heads. The results are presented in Table 6.

In our experiments, Llama 2 models mostly show comparable performance to BERT and GPT-2. However, the Llama 2 70B model surpasses both BERT and GPT-2 in the Inspec and SemEval2017 datasets, achieving the highest scores in F1@5 across all datasets. These superior performances in F1@5 are also observed in our experiments with BERT and GPT-2, indicating a consistent trend across different PLM-based models. Importantly, among the Llama 2 models, there is a clear trend that an increase in the number of parameters leads to better performance. This observation supports the prevalent notion in LLM research that increasing model size can lead to improved performance.

## 7 Related Work

Traditional unsupervised keyphrase extraction (UKE) approaches broadly divided into two categories: statistical-based and graph-based methods. Statistical approaches, like TF-IDF (Jones, 2004) and YAKE (Campos et al., 2018), use information such as word frequency and distribution. Graph-based methods, like TextRank (Mihalcea and Tarau, 2004), SingleRank (Wan and Xiao, 2008), TopicRank (Bougouin et al., 2013), and PositionRank (Florescu and Caragea, 2017), represent a document as a graph and rank phrases based on the graph's characteristics.

Traditional unsupervised keyphrase extraction approaches that rely solely on surface-level features often fail to extract semantically significant

---
[6] https://huggingface.co/meta-llama/Llama-2-7b
[7] https://huggingface.co/meta-llama/Llama-2-13b
[8] https://huggingface.co/meta-llama/Llama-2-70b

keyphrases. To address these limitations, methods utilizing embeddings emerged. EmbedRank (Bennani-Smires et al., 2018) is an example that employs static embeddings such as Sent2Vec and Doc2Vec. Subsequently, a new trend in keyphrase extraction emerged with the advent of PLMs. SIFRank (Sun et al., 2020) utilizes ELMo, a PLM capable of generating contextual embeddings.

More recently, transformer-based models like BERT have been widely adopted. AttentionRank (Ding and Luo, 2021) applies BERT's self-attention and cross-attention mechanisms to keyphrase identification, examining the relevance of a candidate keyphrase within its sentence and the entire document. Masked Document Embedding Rank (MDERank) (Zhang et al., 2022) introduces a masking strategy, ranking keyphrases by comparing the similarity of the source and masked document embeddings. This approach benefits from a customized BERT model trained via a self-supervised contrastive learning method. JointGL (Liang et al., 2021) merges local and global contexts for keyphrase extraction, creating a graph structure to capture local information and calculating semantic similarity for a global perspective.

## 8 Conclusion

In this paper, we present a new perspective on keyphrase extraction that moves away from the conventional embedding-based approach and solely leverages the information from the self-attention map of PLM. The proposed SAMRank extracts keyphrases based on a combination of the global attention score, which identifies tokens that attract attention within a document, and the proportional attention score, which finds tokens deeply associated with important tokens. Experimental results on three representative keyphrase extraction datasets demonstrate that SAMRank consistently outperforms embedding-based models, improving F1@5 and F1@10 scores by approximately 11.5% and 3% on average, respectively.

These results suggest that some heads in PLMs, particularly BERT and GPT-2, could be specialized in capturing keyphrases. We also demonstrate that the self-attention map of these heads can provide more useful information for keyphrase extraction than embeddings. Future work could potentially utilize the SAMRank approach to enhance the interpretability by thoroughly analyzing the specific roles and functionality of each head in transformer-based PLMs, beyond the scope of keyphrase extraction.

## Limitations

SAMRank utilizes just a single self-attention map (SAM) from one out of 144 heads. As evidenced by the results in the Appendix, certain heads consistently exhibit high performance across all datasets, but the best performing head does not always remain the same. Moreover, a trade-off in performance is observed among the heads. Some heads demonstrate high performance on F1@5 but low on F1@15, while others show the opposite pattern. This variability indicates that the optimal head selection may vary depending on the type and length of the text and the specific evaluation metric, necessitating additional human exploration for optimal results. Therefore, rather than solely relying on a single self-attention map, combining outputs from multiple high-performing heads could be expected to yield not only more stable, but also more generalized performance applicable to various types of text. The method of combining multiple SAMs would make a good topic for future research.

Additionally, when processing long documents, we divide the text into segments with equal token counts, extract keyphrases from each segment, and then integrate the results to draw out the final keyphrases. The motivation behind this approach of equal segmentation is to reveal the importance of tokens under identical conditions across all segments. However, this method can lead to a loss of context from the original long text, potentially reducing accuracy. Thus, applying the SAMRank approach to models that can handle more extended token inputs, such as BigBird (Zaheer et al., 2020) or Longformer (Beltagy et al., 2020), could be an interesting topic for future research.

## Acknowledgements

This research was supported by the MSIT(Ministry of Science and ICT), Korea, under the ICAN(ICT Challenge and Advanced Network of HRD) support program(IITP-2023-RS-2023-00260175) supervised by the IITP(Institute for Information & Communications Technology Planning & Evaluation).

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

# A Performance of All Heads in BERT and GPT-2

We evaluate the keyphrase extraction performance of all the heads in BERT and GPT-2 for three datasets (Inspec, SemEval2010, SemEval2017) using the SAMRank. Figures 3 and 4 represent the F1@15 score results of the heads of each layer of BERT and GPT-2 respectively in a graph form. The horizontal axis of the graph signifies the layer order, and the heads from 1 to 12 in each layer are denoted by different symbols. The graph at the top shows the performance for the Inspec dataset, the one in the middle for SemEval2010, and the one at the bottom for the SemEval2017 dataset.

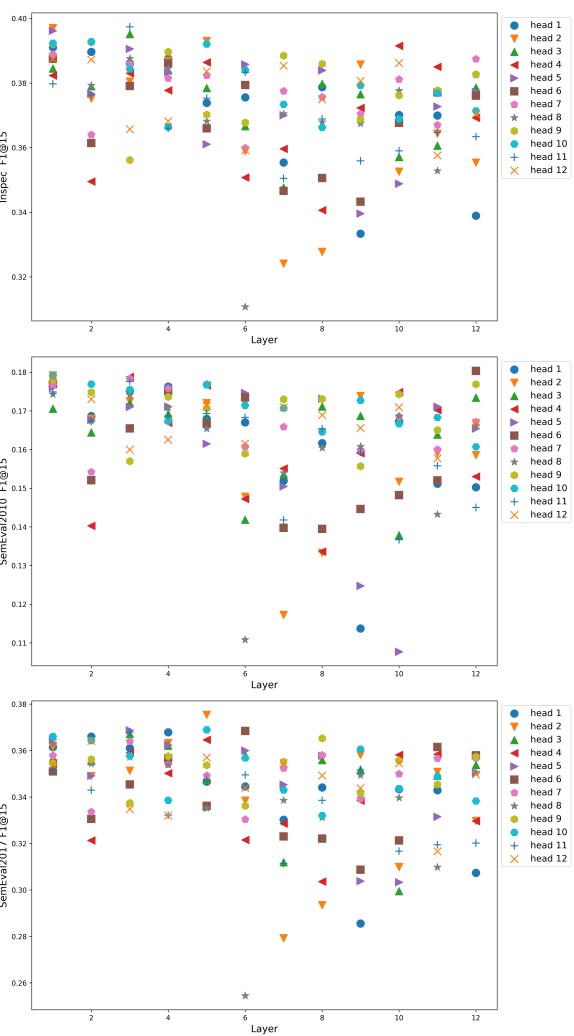

Figure 3: Performance (F1@15) of BERT's heads in each layer for three datasets.

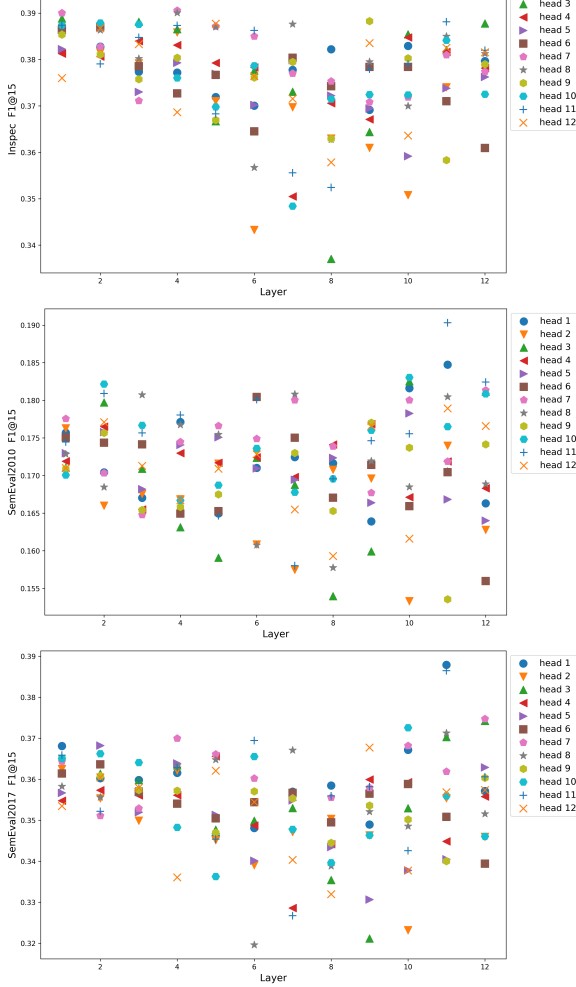

Figure 4: Performance (F1@15) of GPT-2's heads in each layer for three datasets.

## B  Visualization of Actual Self-Attention Maps and Token-level Scores

Figures 5 and 6 visualize the self-attention map (SAM) obtained by inputting example documents from SemEval2017, and the token-level scores calculated through the SAM. The SAM in Figure 5 is extracted from the 5-2 (layer-head) of BERT, which shows the highest performance in SemEval2017. The SAM in Figure 6 is extracted from the 11-1 head of GPT-2, which shows the highest performance in SemEval2017. The figures below the SAM visualize the final token-level scores obtained by combining the proposed global attention score and proportional attention score in the paper.

| Model | Krapivin | | |
|---|---|---|---|
| | F1@5 | F1@10 | F1@15 |
| TextRank | 6.04 | 9.43 | 9.95 |
| SingleRank | 8.12 | 10.53 | 10.42 |
| TopicRnak | 8.94 | 9.01 | 8.30 |
| MultipartiteRank | 9.29 | 9.35 | 9.16 |
| YAKE | 8.09 | 9.35 | 11.05 |
| EmbedRank(Sent2Vec)+MMR | 8.44 | 10.47 | 10.71 |
| SIFRank(ELMo) | 1.62 | 2.52 | 3.00 |
| EmbedRank(BERT) | 4.05 | 6.60 | 7.84 |
| MDERank(BERT) | 11.78 | 12.93 | 12.58 |
| MDERank(KPEBERT$_{ab}$) | 12.91 | 14.36 | 13.58 |
| MDERank(KPEBERT$_{re}$) | 12.35 | 14.31 | 13.31 |
| **SAMRank (BERT)** | **16.35** | **15.91** | **14.52** |
| **SAMRank (GPT-2)** | **17.49** | **16.46** | **14.92** |

Table 7: Performance of SAMRank on the Krapivin dataset for very long documents.

## C  SAMRank on Long Documents

Transformer-based Pre-trained Language Models (PLMs), which are inherently based on attention mechanisms, have demonstrated outstanding performance. However, the inherent characteristic of attention mechanisms is that their computational cost increases with the length of the input. Models like BERT and GPT-2 have a token input limit, typically set at 512 or 1024 tokens. To address this limitation, SAMRank divides longer documents into smaller segments. Each segment is processed separately, and the scores from each segment are aggregated to derive the final keyphrases. To validate the effectiveness of this approach on very long documents, we conduct experiments with this method on the Krapivin dataset (Krapivin and Marchese, 2009), comprised of scientific full papers that average about 8,500 words. The experimental results are presented in Table 7. The performance of the baselines was referenced from the MDERank paper (Zhang et al., 2022).

SAMRank, when utilizing BERT and GPT-2, achieves the highest F1 scores across all three metrics, demonstrating its effectiveness in handling very long documents. This suggests that even when only a part of the entire document is inputted and the context is compromised, attention weights still strongly focuses on the keyphrases. Moreover, the F1@5 shows exceptionally high performance compared to other metrics, indicating a tendency for strong attention concentration on a few tokens within a single attention map.

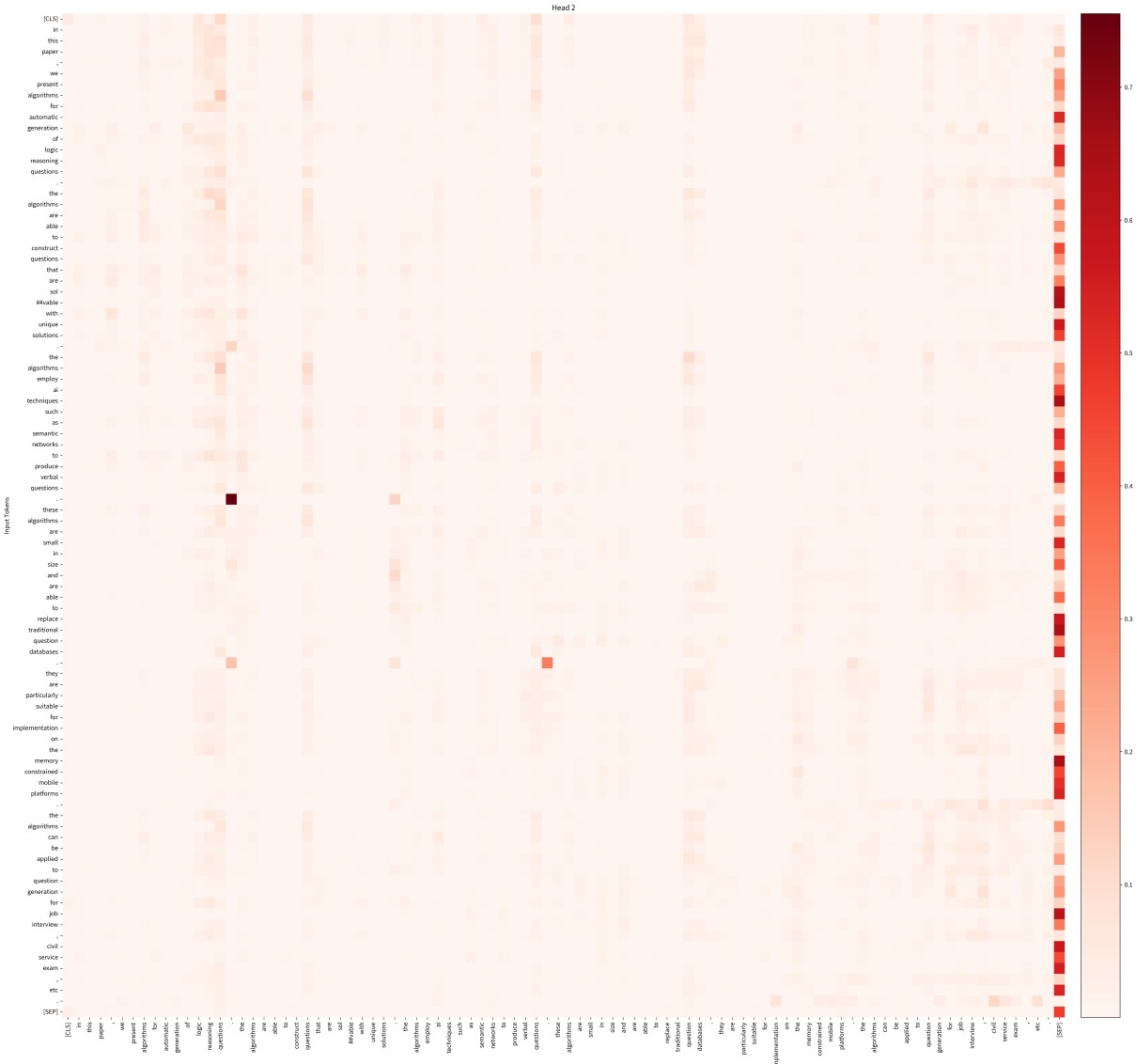

Head 2

[CLS] in this paper , we present algorithms for automatic generation of logic reasoning questions . the algorithms are able to construct questions that are sol ##vable with unique solutions . the algorithms employ ai techniques such as semantic networks to produce verbal questions . these algorithms are small in size and are able to replace traditional question databases . they are particularly suitable for implementation on the memory constrained mobile platforms . the algorithms can be applied to question generation for job interview , civil service exam , etc . [SEP]

Figure 5: Visualization of example self-attention map extracted from BERT's head (5-2) and token-level scores calculated by combining global attention score and proportional attention score.

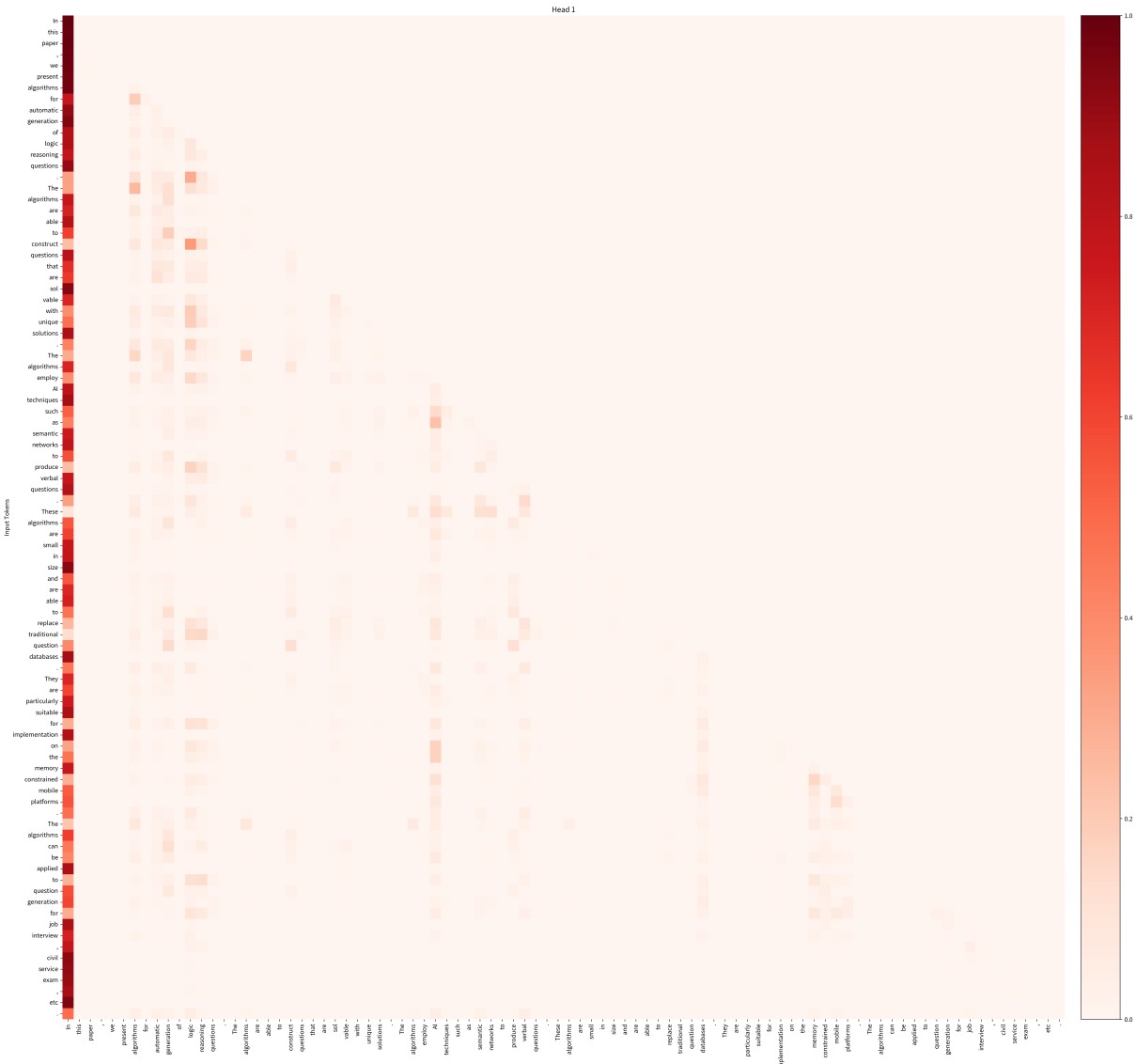

In this paper, we present algorithms for automatic generation of logic reasoning questions. The algorithms are able to construct questions that are solvable with unique solutions. The algorithms employ AI techniques such as semantic networks to produce verbal questions. These algorithms are small in size and are able to replace traditional question databases. They are particularly suitable for implementation on the memory constrained mobile platforms. The algorithms can be applied to question generation for job interview, civil service exam, etc.

Figure 6: Visualization of example self-attention map extracted from GPT-2's head (11-1) and token-level scores calculated by combining global attention score and proportional attention score.