# OpenReview forum: "SAMRank: Unsupervised Keyphrase Extraction using Self-Attention Map in BERT and GPT-2"
_EMNLP/2023/Conference — EMNLP 2023 Main_

### Official Review · Reviewer_Tn1j · 2023-07-25

**Soundness:** 4

**Excitement:**

4: Strong: This paper deepens the understanding of some phenomenon or lowers the barriers to an existing research direction.

**Paper Topic And Main Contributions:**

This paper presents a study on unsupervised keyphrase extraction, with the central motivation being how to leverage the weight information from the attention matrices in pre-trained models during the text representation process to ascertain the importance of different tokens. The experimental section demonstrates that the proposed approach based on attention matrices outperforms methods based on embedding representation and similarity measures, thereby providing evidence for the efficacy of the SAMRank algorithm proposed in this paper. The experiments conducted in this study are comprehensive and effectively establish that the observed improvements stem from the proposed SAMRank algorithm.

**Questions For The Authors:**

Suggestion:
1. Have you contemplated investigating the disparities between attention mechanisms in large-scale language models (e.g., 7B, 13B, ~100B) and their smaller-scale counterparts?

**Reasons To Accept:**

1 This paper proposes an intriguing approach that distinguishes itself from conventional representation-based unsupervised keyphrase extraction algorithms. Instead, this study utilizes the values from the attention matrices of BERT and GPT to quantify the importance of candidate keyword phrases, resulting in a good performance.

2 The experimental section of this paper is remarkably comprehensive, and the experimental results are highly impressive. These experiments thoroughly validate the effectiveness of the proposed method.

**Reasons To Reject:**

1. While the method proposed in this paper is the first to utilize attention matrices for keyphrase extraction, there is a lack of sufficient research on how to select the appropriate layers and heads of attention matrices.

2. I have a potentially valid concern regarding the lack of exploration of the method proposed in this paper in relation to large models at the current stage.

**Reproducibility:**

5: Could easily reproduce the results.

**Reviewer Confidence:**

5: Positive that my evaluation is correct. I read the paper very carefully and I am very familiar with related work.

---

> ### Author Rebuttal · Authors · 2023-08-29
>
> We would like to extend our sincerest thanks for your comprehensive and thoughtful review of our paper. Your insightful concerns have been invaluable in helping us further refine and improve our work.
> \
> &nbsp;
>
> ### **The disparities between attention mechanisms in LLM and their smaller-scale counterparts**
>
> In our study, we observed promising results for keyphrase extraction using both encoder-based BERT and decoder-based GPT-2. This led us to wonder about the efficacy of newer models like GPT-3.5 and GPT-4, which were not open-sourced at the time of our research. Fortunately, the recent release of the open-source Llama2 models allowed us to extend our experiments. We used the Llama2 7B and 13B models, which are significantly larger than BERT and GPT-2, containing up to 1,600 attention heads (about 11 times BERT and GPT-2).
>
> SE2010: SemEval2010
>
> SE2017: SemEVal2017
> |                | Inspec |      |      | SE2010 |      |      | SE2017 |      |      |
> |----------------|--------|------|------|-------------|------|------|-------------|------|------|
> | Model          | F1@5   | F1@10| F1@15| F1@5        | F1@10| F1@15| F1@5        | F1@10| F1@15|
> | SAMRank(BERT)  | 33.96  | 39.35| 39.73| 15.28       | 18.36| 18.03| 24.08       | 33.40| 37.53|
> | SAMRank(GPT-2) | 33.92  | 39.44| 39.72| 15.88       | 19.49 | 19.03| 24.80   | 34.75| 38.78|
> | **SAMRank(Llama7B)** | 34.26 | 39.04| 39.20| 16.69      | 18.65| 18.50| 23.21       | 33.37| 37.62|
> | **SAMRank(Llama13B)** | 34.06| 39.74 | 39.50| 16.11       | 18.46| 19.03| 23.93       | 33.96| 38.04|
>
> Our experiments with Llama2 generally mirrored the performance of BERT and GPT-2, with some instances of improved performance. This similarity in performance allows us to hypothesize that there may not be significant disparities in attention mechanisms between smaller models and LLMs when it comes to specific heads focusing on keyphrases.
>
> - We observed an improvement in the F1@5 metric in the SemEval2010 dataset when using Llama2. We hypothesize that **this is because LLMs like Llama2 are capable of processing a larger number of tokens, thus allowing for a more contextually complete Self-Attention Map** to be generated for keyphrase extraction.
>
> - Our additional experiments show that the size of the language model doesn't significantly affect keyphrase extraction performance.  **We cautiously speculate that while larger models can process language from more diverse perspectives, it doesn't necessarily make individual heads more effective.**
>
> - We also intended to conduct experiments on Llama2 70B, but due to time constraints, we were unable to include them. **We will complete the experiments and add the performance results for Llama2 70B in the revised version of the paper.**
>
> To offer deeper insights into Large Language Models (LLMs), we plan to include a comprehensive analysis of performance across various heads in Appendix A and visualizations of Self-Attention Maps in Appendix B of our paper.
> \
> &nbsp;
>
> In closing, we appreciate your time, effort, and the expertise that went into your constructive review. We hope that additional experiments have addressed your insightful concerns.

---

### Official Review · Reviewer_4f1S · 2023-08-03

**Soundness:** 2

**Excitement:**

3: Ambivalent: It has merits (e.g., it reports state-of-the-art results, the idea is nice), but there are key weaknesses (e.g., it describes incremental work), and it can significantly benefit from another round of revision. However, I won't object to accepting it if my co-reviewers champion it.

**Paper Topic And Main Contributions:**

Keyphrase prediction is an interesting task in NLP. The paper proposes an unsupervised keyphrase extraction model, which uses only a self-attention map in BERT and GPT-2l to compute the importance of the candidate phrases.


**Questions For The Authors:**





**Reasons To Accept:**

1. The paper contributes a new idea, i.e.,  in an unsupervised model, a self-attention map in BERT and GPT-2 is used to extract keyphrases.

2. The paper is detailed in each step and the source code has been released.


**Reasons To Reject:**

1. The technological novelty of this paper is a bit limited as some keyphrase prediction methods have combined the transformer-based models with the traditional unsupervised method.

2. The comparable methods only include some unsupervised keyphrase extraction models. The latest neural keyphrase generation models, such as CopyRNN (Meng et al., 2017) and SEG-Net (Ahmad et al., 2021), should be included to evaluate the proposed model.

3. More datasets, such as KP20k (Meng et al., 2017) and Krapivin (Krapivin et al., 2009), should be used to evaluate the proposed model.

**Reproducibility:**

5: Could easily reproduce the results.

**Reviewer Confidence:**

5: Positive that my evaluation is correct. I read the paper very carefully and I am very familiar with related work.

---

> ### Author Rebuttal · Authors · 2023-08-29
>
> We greatly appreciate the time and effort you've invested in reviewing our paper. Your detailed feedback provides invaluable insights for improving our work.
> \
> &nbsp;
> ### **Technological Novelty**
>
> We appreciate your observation that existing methods already combine transformer-based models with traditional unsupervised techniques. Our baseline models, such as AttentionRank, MDERank, and JointGL, indeed leverage transformer-based models. However, it's important to underline that SAMRank takes a fundamentally different approach. **While existing models rely on the embeddings generated by transformers**, our work exploits solely the Self-Attention Maps of specific transformer heads.
>
> We've discussed in the Introduction section the potential limitations of embedding-based methods, citing research that shows the cosine similarity between arbitrary word embeddings from upper transformer layers approaches 1. **Embedding-based methods may not be suitable for tasks requiring ranking by similarity, hence SAMRank bypasses this limitation by focusing solely on Self-Attention Map.**
>
> SAMRank is rooted in an intriguing behavior observed in specific transformer heads, which closely aligns with how humans often underline or highlight keyphrases while reading. This behavior is observed in specific heads of Transformer models like BERT and GPT-2, which focus attention on semantically important keyphrases. In Appendix B's Figure 5 (BERT’s Self-Attention Map), for instance, the majority of tokens in the text are shown to assign higher attention weights to the phrase 'logic reasoning question'. This pattern is further substantiated in Figure 6, which visualizes attention allocation in the 11th layer's 1st head of GPT-2, highlighting attention weights towards tokens such as 'algorithms,' 'automatic generation,' 'logic reasoning questions,' and 'AI.'
>
> **As the first study to leverage this intriguing attribute inherent in Transformers, our method is significantly different** from existing transformer-based methods that aim to improve performance through embedding similarities.
> \
> &nbsp;
> ### **Comparing with Supervised Keyphrase Generation Models**
>
> Your suggestion to include keyphrase generation models like CopyRNN and SEG-Net as baselines is constructive. However, these models are **supervised**, meaning they are trained on labeled keyphrase data. In contrast, **SAMRank is unsupervised and does not rely on gold keyphrases or training.**
>
> Moreover, the very nature of generative models allows them to predict both present and absent keyphrases, making a direct comparison challenging. Extractive models like SAMRank are designed to predict 'present keyphrases,' which are phrases directly appearing in the source text. In contrast, generative models are trained not only on present keyphrases label but also on absent keyphrases label, enabling them to predict both. As a result, the performance of generative models is often separately evaluated for 'present' and 'absent' keyphrases. This is different from extractive models, which are evaluated based on a ground truth that includes both present and absent keyphrases. Even if we were to compare only the present keyphrase prediction performance of generative models, **the difference in the ground truth (gold keyphrases) used for evaluation makes a direct comparison between the two types of models less straightforward.**
>
> It's worth noting that recent EMNLP or ACL papers on unsupervised keyphrase extraction, such as AttentionRank (Ding & Luo, 2021), JointGL (Liang et al., 2021) and MDERank (Zhang et al., 2022),  have also chosen to compare only with other unsupervised extraction models. We were following this research trend for a methodologically consistent and fair comparison with existing approaches. But we **will include a discussion on generative models like CopyRNN and SEG-Net in the Related Work section to provide a comprehensive view.**
> \
> &nbsp;
> ### **Additional Dataset**
> | Dataset                 | Krapivin                |
> |-------------------------|-------------------------|
> | #Doc                    | 460                     |
> | Avg. #words             | 9057                    |
> | Avg. #Sents             | 345                     |
> | Avg. #Keys              | 6                       |
> | #unigram                | 17.79%                  |
> | #bigram                 | 62.21%                  |
> | #trigram                | 16.35%                  |
>
> Results on Krapivin dataset:
> | Model                     | F1@5  | F1@10 | F1@15  |
> |---------------------------|-------|-------|--------|
> | TextRank                  | 6.04  | 9.43  | 9.95   |
> | SingleRank                | 8.12  | 10.53 | 10.42  |
> | TopicRnak                 | 8.94  | 9.01  | 8.30   |
> | MultipartiteRank          | 9.29  | 9.35  | 9.16   |
> | YAKE                      | 8.09  | 9.35  | 11.05  |
> | EmbedRank(Sent2Vec)+MMR   | 8.44  | 10.47 | 10.71  |
> | SIFRank(ELMo)             | 1.62  | 2.52  | 3.00   |
> | EmbedRank(BERT)           | 4.05  | 6.60  | 7.84   |
> | JointGL                   | 5.47  | 6.12  | 6.60   |
> | MDERank(BERT)             | 11.78 | 12.93 | 12.58  |
> | MDERank(KPEBERTab)        | 12.91 | 14.36 | 13.58  |
> | MDERank(KPEBERTre)        | 12.35 | 14.31 | 13.31  |
> | **SAMRank(BERT)**         | **16.31** | **15.20** | **13.87**  |
> | **SAMRank(GPT-2)**        | **16.70** | **15.97** | **14.63**  |
>
> We've evaluated SAMRank on the Krapivin dataset as you suggested and achieved state-of-the-art performance, demonstrating its effectiveness even on very long documents. **These results will be added to Table 3.**
> \
> &nbsp;
> ### **SAMRank with LLM (Llama2)**
> SE2010: SemEval2010
>
> SE2017: SemEval2017
> |                | Inspec |      |      | SE2010 |      |      | SE2017 |      |      |
> |----------------|--------|------|------|-------------|------|------|-------------|------|------|
> | Model          | F1@5   | F1@10| F1@15| F1@5        | F1@10| F1@15| F1@5        | F1@10| F1@15|
> | SAMRank(BERT)  | 33.96  | 39.35| 39.73| 15.28       | 18.36| 18.03| 24.08       | 33.40| 37.53|
> | SAMRank(GPT-2) | 33.92  | 39.44| 39.72| 15.88       | 19.49 | 19.03| 24.80   | 34.75| 38.78|
> | **SAMRank(Llama7B)** | 34.26 | 39.04| 39.20| 16.69      | 18.65| 18.50| 23.21       | 33.37| 37.62|
> | **SAMRank(Llama13B)** | 34.06| 39.74 | 39.50| 16.11       | 18.46| 19.03| 23.93       | 33.96| 38.04|
>
> Additionally, we conducted further experiments using the recently-released Llama2 model to assess SAMRank's performance across different transformer architectures. Preliminary results indicate that the model outperforms the proposed baselines, further **supporting the generalizability of our approach**.
>
> We also intended to conduct experiments on Llama2 70B, but due to time constraints, we were unable to include them. We will complete the experiments and add the performance results for Llama2 70B in the revised version of the paper.
> \
> &nbsp;
>
> We deeply appreciate your invaluable insights and thoughtful suggestions. We hope that our responses and additional experiments address your concerns.

---

### Official Review · Reviewer_aUyH · 2023-08-03

**Soundness:** 3

**Excitement:**

3: Ambivalent: It has merits (e.g., it reports state-of-the-art results, the idea is nice), but there are key weaknesses (e.g., it describes incremental work), and it can significantly benefit from another round of revision. However, I won't object to accepting it if my co-reviewers champion it.

**Missing References:**

Krapivin M, Autaeu A, Marchese M. Large dataset for keyphrases extraction. Technical Report, University of Trento, 2008

**Paper Topic And Main Contributions:**

The topic of this paper is about approaches for data- and compute efficiency.

The main contributions of this paper are as follows:

(1)This paper presents a SAMRank model for keyphrase extraction that solely leverages the information from the self-attention map of PLM.

 (2)The proposed extracts keyphrases based on a combination of the global attention score, which identifies tokens that attract attention within a document, and the proportional attention score, which finds tokens deeply associated with important tokens.

 (3)Experimental results demonstrate that SAMRank is effective on three representative keyphrase extraction datasets.


**Questions For The Authors:**

(1)In ，In Section 3.5, Bert is in token, what is the boundary of the phrase given? Please give a detailed explanation.


(2) This proposed method only uses the Self-Attention Map Extraction and  Attention Score, and its performance can achieve better results. Please give a detailed explanation.
(3) Add some data sets to evaluate the proposed model,  such as Krapivin (Krapivin M, Autaeu A, Marchese M. Large dataset for keyphrases extraction. Technical Report, University of Trento, 2008.).


**Reasons To Accept:**

（1）The proposed method seems to show fairly good results on three representative keyphrase extraction datasets.

 (2) The  source code has been released.

**Reasons To Reject:**

(1)The innovation of the proposed method is limited using self-attention map.

(2) This proposed method only uses the Self-Attention Map Extraction and  Attention Score, and its performance can achieve better results. Please give a detailed explanation

(3) Add some data sets to evaluate the proposed model,  such as Krapivin (Krapivin M, Autaeu A, Marchese M. Large dataset for keyphrases extraction. Technical Report, University of Trento, 2008.).


**Reproducibility:**

5: Could easily reproduce the results.

**Reviewer Confidence:**

4: Quite sure. I tried to check the important points carefully. It's unlikely, though conceivable, that I missed something that should affect my ratings.

---

> ### Author Rebuttal · Authors · 2023-08-29
>
> Thank you for your thorough review and insightful suggestions on our paper, which contribute to enhancing the quality of our work.
> \
> &nbsp;
> ### **Question 1 : What is the boundary of the phrase given?**
>
> As you rightly pointed out, both BERT and GPT-2 provide self-attention maps at the token level. We calculate the scores for each token in the text and subsequently sum the scores of tokens that constitute the candidate phrases. For better clarity, we refer to Figure 5 in Appendix B, where a self-attention map from one of BERT's heads is visualized. The score of a phrase, like "logic reasoning questions," is determined by summing the scores of its constituent tokens, specifically the 13th ('logic'), 14th ('reasoning'), and 15th ('questions') tokens in the text. While the word 'questions' may appear multiple times in the text, only the 15th token's score will contribute to the score of the phrase 'logic reasoning questions'.
>
> To implement this, we first remove tokenizer-specific special characters like '##' (BERT) and 'Ġ’ (GPT-2). Next, **we store the start and end token indices of each candidate phrase within the text.** If a certain phrase appears multiple times in the text, we store the indices for each occurrence. Only the tokens that exactly match the candidate phrase, without stemming, will contribute to its final score. **We plan to add more detailed information in Section 3 of the revised manuscript.**
> \
> &nbsp;
> ### **Question 2 : Detailed explanation on why SAMRank achieves better results**
>
> The performance of SAMRank is not solely attributed to our proposed methodology but also to inherent characteristics commonly observed in Transformer-based pre-trained language models like BERT and GPT-2. **Our model builds upon the observation that certain heads in these Transformer models are more inclined to focus on semantically important keyphrases.** Through qualitative analysis conducted on the SemEval2017 dataset, as visualized in Appendix B (Figure 5), we observed this attention behavior. Furthermore, we extended this qualitative evaluation to GPT-2 and found similar tendencies (Figure 6). This behavior is not confined to specific models; In additional experiments, we confirmed its universality by applying SAMRank to a recently released large language model, Llama2, achieving comparable results.
> \
> &nbsp;
>
> SE2010: SemEval2010
>
> SE2017: SemEval2017
> |                | Inspec |      |      | SE2010 |      |      | SE2017 |      |      |
> |----------------|--------|------|------|-------------|------|------|-------------|------|------|
> | Model          | F1@5   | F1@10| F1@15| F1@5        | F1@10| F1@15| F1@5        | F1@10| F1@15|
> | SAMRank(BERT)  | 33.96  | 39.35| 39.73| 15.28       | 18.36| 18.03| 24.08       | 33.40| 37.53|
> | SAMRank(GPT-2) | 33.92  | 39.44| 39.72| 15.88       | 19.49 | 19.03| 24.80   | 34.75| 38.78|
> | **SAMRank(Llama7B)** | 34.26 | 39.04| 39.20| 16.69      | 18.65| 18.50| 23.21       | 33.37| 37.62|
> | **SAMRank(Llama13B)** | 34.06| 39.74 | 39.50| 16.11       | 18.46| 19.03| 23.93       | 33.96| 38.04|
>
> We also intended to conduct experiments on Llama2 70B, but due to time constraints, we were unable to include them. We will complete the experiments and add the performance results for Llama2 70B in the revised version of the paper.
>
> In summary, the strength of SAMRank lies not just in its algorithmic design **but also in leveraging the intrinsic characteristics of Transformer models that inherently focus on semantically important keyphrases.** This is a behavior confirmed across different models and datasets. Therefore, the effectiveness of SAMRank is, in part, a reflection of the pre-trained models' built-in capability to give attention to keyphrases.
>
> We are grateful for the request for additional clarification on this matter. The current manuscript may not fully explain why SAMRank excels with just the use of self-attention maps; however, we hope to address this gap in the next manuscript submission.
> \
> &nbsp;
> ### **Question 3 : Additional experimental results on Krapivin**
>
> We are grateful for the suggestion to evaluate our model on the Krapivin dataset. Upon conducting additional experiments, **our model achieved state-of-the-art results. These statistics and results will be included in the revised paper.**
>
>
> | Dataset                 | Krapivin                |
> |-------------------------|-------------------------|
> | #Doc                    | 460                     |
> | Avg. #words             | 9057                    |
> | Avg. #Sents             | 345                     |
> | Avg. #Keys              | 6                       |
> | #unigram                | 17.79%                  |
> | #bigram                 | 62.21%                  |
> | #trigram                | 16.35%                  |
>
> Results on Krapivin dataset:
> | Model                     | F1@5  | F1@10 | F1@15  |
> |---------------------------|-------|-------|--------|
> | TextRank                  | 6.04  | 9.43  | 9.95   |
> | SingleRank                | 8.12  | 10.53 | 10.42  |
> | TopicRnak                 | 8.94  | 9.01  | 8.30   |
> | MultipartiteRank          | 9.29  | 9.35  | 9.16   |
> | YAKE                      | 8.09  | 9.35  | 11.05  |
> | EmbedRank(Sent2Vec)+MMR   | 8.44  | 10.47 | 10.71  |
> | SIFRank(ELMo)             | 1.62  | 2.52  | 3.00   |
> | EmbedRank(BERT)           | 4.05  | 6.60  | 7.84   |
> | JointGL                   | 5.47  | 6.12  | 6.60   |
> | MDERank(BERT)             | 11.78 | 12.93 | 12.58  |
> | MDERank(KPEBERTab)        | 12.91 | 14.36 | 13.58  |
> | MDERank(KPEBERTre)        | 12.35 | 14.31 | 13.31  |
> | **SAMRank(BERT)**         | **16.31** | **15.20** | **13.87**  |
> | **SAMRank(GPT-2)**        | **16.70** | **15.97** | **14.63**  |
>
> We appreciate your insights and the time you have invested in reviewing our paper. We hope to submit a revised paper that reflects your valuable feedback.

---

### Official Review · Reviewer_G3r8 · 2023-08-03

**Soundness:** 4

**Excitement:**

4: Strong: This paper deepens the understanding of some phenomenon or lowers the barriers to an existing research direction.

**Paper Topic And Main Contributions:**

This paper presents SAMRank, a novel unsupervised keyphrase extraction method that utilizes a self-attention map from a pre-trained language model (PLM) to determine phrase importance. The motivation for this approach is twofold. Firstly, contextualized representations, as observed in PLMs like ELMo, BERT, and GPT-2, often possess anisotropy, making them less suitable for similarity-based tasks. Secondly, previous studies have indicated that certain heads in transformer-based PLMs, such as BERT, specialize in specific tasks, thereby highlighting the potential of using self-attention maps for keyphrase extraction.

The authors' description of the motivation for their model design is detailed and convincing, making the paper easy to comprehend and follow. The experimental section includes a thorough comparison of SAMRank with baseline approaches on three public datasets. The results obtained from these experiments demonstrate the effectiveness of SAMRank in keyphrase extraction.

**Questions For The Authors:**

Could you please elaborate further on the results presented in Table 3?

**Reasons To Accept:**

1. This paper introduces SAMRank, a novel unsupervised keyphrase extraction method that leverages the self-attention map of a pre-trained language model (PLM) to determine the importance of phrases. The proposed approach is technical sound.

2. The motivation behind this work is both clear and convincing. Also, the paper is well-organized and easy to follow.

3. The experimental results show that SAMRank outperforms most embedding-based models on both long and short documents and demonstrating that it is possible to use only a self-attention map for keyphrase extraction without relying on embeddings.

**Reasons To Reject:**

On some datasets, the results of SAMBank are not significantly better than the baseline method.

**Reproducibility:**

5: Could easily reproduce the results.

**Reviewer Confidence:**

3: Pretty sure, but there's a chance I missed something. Although I have a good feel for this area in general, I did not carefully check the paper's details, e.g., the math, experimental design, or novelty.

---

> ### Author Rebuttal · Authors · 2023-08-29
>
> We appreciate the opportunity to respond to your insightful questions. Your expertise in this area is invaluable to us, and we take your comments very seriously as we strive to improve the quality of our work.
> \
> &nbsp;
>
> ###  **Detailed description of Table 3**
>
> As observed in Table 3, the **F1@5 performance of SAMRank is significantly higher across all datasets compared to F1@10 and F1@15.** This higher score at F1@5 can be attributed to an interesting phenomenon observed in specific heads of Transformer-based Pretrained Language Models (PLMs), which is further detailed in the Self-Attention Maps presented in Appendix B.
>
> In Figures 5 (BERT’s Self-Attention Map) of Appendix B, exhibit vertical lines of darker colors corresponding to the tokens that form the phrase 'logic reasoning questions’. This implies that the majority of tokens in the text are assigning relatively higher attention weights to each token in the phrase 'logic reasoning questions.' Particularly, Figure 6, which represents the Self-Attention Map from the 11th layer's 1st head of GPT-2, reveals that a substantial number of attention weights are allocated to tokens such as 'algorithms,' 'automatic generation', 'logic reasoning questions' and 'AI’.
>
> Therefore, it can be inferred that SAMRank's notably **high performance in F1@5 is due to the intriguing phenomenon where attention is heavily focused on the tokens that make up keyphrases in specific heads of the Transformer.**
> \
> &nbsp;
> ### **Additional experimental results on Krapivin**
>
> Moreover, we have conducted additional experiments on the Krapivin dataset and observed similar trends. Despite the Krapivin documents being quite long, averaging around 9,000 words, they contain relatively few ground truth keyphrases, averaging about 6. **SAMRank achieves state-of-the-art performance in F1@5, F1@10, and F1@15 metrics on this dataset (Krapivin) as well, significantly outperforming baseline models, especially in F1@5.**
>
> | Dataset                 | Krapivin                |
> |-------------------------|-------------------------|
> | #Doc                    | 460                     |
> | Avg. #words             | 9057                    |
> | Avg. #Sents             | 345                     |
> | Avg. #Keys              | 6                       |
> | #unigram                | 17.79%                  |
> | #bigram                 | 62.21%                  |
> | #trigram                | 16.35%                  |
>
>
>
> Results on Krapivin dataset:
> | Model                     | F1@5  | F1@10 | F1@15  |
> |---------------------------|-------|-------|--------|
> | TextRank                  | 6.04  | 9.43  | 9.95   |
> | SingleRank                | 8.12  | 10.53 | 10.42  |
> | TopicRnak                 | 8.94  | 9.01  | 8.30   |
> | MultipartiteRank          | 9.29  | 9.35  | 9.16   |
> | YAKE                      | 8.09  | 9.35  | 11.05  |
> | EmbedRank(Sent2Vec)+MMR   | 8.44  | 10.47 | 10.71  |
> | SIFRank(ELMo)             | 1.62  | 2.52  | 3.00   |
> | EmbedRank(BERT)           | 4.05  | 6.60  | 7.84   |
> | JointGL                   | 5.47  | 6.12  | 6.60   |
> | MDERank(BERT)             | 11.78 | 12.93 | 12.58  |
> | MDERank(KPEBERTab)        | 12.91 | 14.36 | 13.58  |
> | MDERank(KPEBERTre)        | 12.35 | 14.31 | 13.31  |
> | **SAMRank (BERT)**         | **16.31** | **15.20** | **13.87**  |
> | **SAMRank (GPT-2)**        | **16.70** | **15.97** | **14.63**  |
>
> We plan to enhance the manuscript by adding the additional experiment result for the Krapivin dataset to Table 3, and we also intend to provide a detailed explanation of Table 3 in the 4.3 Results section as you suggested.
> \
> &nbsp;
>
> We are grateful for your time and thorough review, which are instrumental in enhancing the quality of our manuscript.

---

### Meta-Review · Area_Chair_ckkA · 2023-09-21

**Recommendation:** 5

**Metareview:**

This paper introduces a novel unsupervised keyphrase extraction method called SAMRank, which uses the self-attention map of PLMs to determine the importance of phrases. The idea of computing scores based on self-attention map is simple but still novel in this task. More datasets are encouraged.

---

### Decision · Program_Chairs · 2023-10-07

**Decision:**

Accept-Main

**Comment:**

This paper introduces a novel unsupervised keyphrase extraction method called SAMRank, which uses the self-attention map of PLMs to determine the importance of phrases. The idea of computing scores based on self-attention map is simple but still novel in this task. More datasets are encouraged.